# The Pharmaceutical Ability of *Pistacia lentiscus* L. Leaves Essential Oil Against Periodontal Bacteria and *Candida* sp. and Its Anti-Inflammatory Potential

**DOI:** 10.3390/antibiotics9060281

**Published:** 2020-05-26

**Authors:** Egle Milia, Marianna Usai, Barbora Szotáková, Marie Elstnerová, Věra Králová, Guy D’hallewin, Ylenia Spissu, Antonio Barberis, Mauro Marchetti, Antonella Bortone, Vincenzo Campanella, Giorgio Mastandrea, Lenka Langhansová, Sigrun Eick

**Affiliations:** 1Department of Medicine, Surgery and Experimental Science, University of Sassari, Viale San Pietro 43, 07100 Sassari, Italy; 2Department of Chemistry and Pharmacy, University of Sassari, Via Rolando, 07100 Sassari, Italy; dsfusai@uniss.it; 3Faculty of Pharmacy, Charles University, Akademika Heyrovského 1203, 50005 Hradec Králové, Czech Republic; szotakova@faf.cuni.cz (B.S.); elstnerovam@faf.cuni.cz (M.E.); 4Faculty of Medicine, Charles University, Šimkova 870, 50003 Hradec Králové, Czech Republic; kralovav@lfhk.cuni.cz; 5National Research Council-Institute of Sciences of Food Production, Traversa La Crucca 3, Loc. Baldinca, 07100 Sassari, Italy; guy.dhallewin@cnr.it (G.D.); yspissu@uniss.it (Y.S.); antonio.barberis@cnr.it (A.B.); 6National Research Council-Institute of Biomolecular Chemistry, Traversa La Crucca 3, Loc. Baldinca, 07100 Sassari, Italy; mauro@ss.cnr.it; 7Dental Unite, Department of Surgery, Azienda Ospedaliero Universitaria, 07100 Sassari, Italy; antonella.bortone@aousassari.it; 8Department of Clinical and Translational Medicine, University of Rome, Tor Vergata, 00133 Rome, Italy; vincenzo.campanella@uniroma2.it; 9Department of Biomedical Science, University of Sassari, Viale San Pietro 43/C, 07100 Sassari, Italy; 30050744@studenti.uniss.it; 10Institute of Experimental Botany, Czech Academy of Sciences, Rozvojová 263, 16502 Prague, Czech Republic; langhansova@ueb.cas.cz; 11Department of Periodontology, School of Dental Medicine, University of Bern, Freiburgstrasse 7, 3010 Bern, Switzerland

**Keywords:** oral health care products, cyclooxygenase, lipoxygenase, periodontal disease, *Candida albicans*, *Candida glabrata*, medicinal herbs

## Abstract

Background: Given the increasing request for natural pharmacological molecules, this study assessed the antimicrobial capacity of *Pistacia lentiscus* L. essential oil (PLL-EO) obtained from the leaves of wild plants growing in North Sardinia (Italy) toward a wide range of periodontal bacteria and *Candida,* including laboratory and clinical isolates sp., together with its anti-inflammatory activity and safety. Methods: PLL-EO was screened by gas chromatography/mass spectrometry. The minimal inhibitory concentration (MIC) was determined. The anti-inflammatory activity was measured by cyclooxygenase (COX-1/2) and lipoxygenase (LOX) inhibition, while the antioxidant capacity was determined electro-chemically and by the MTT assay. The WST-1 assay was used to ascertain cytotoxicity toward four lines of oral cells. Results: According to the concentrations of terpens, PLL-EO is a pharmacologically-active phytocomplex. MICs against periodontal bacteria ranged between 3.13 and 12.5 µg/ml, while against *Candida* sp. they were between 6.25 and 12.5 µg/mL. Oxidation by COX-1/2 and LOX was inhibited by 80% and 20% µg/mL of the oil, respectively. Antioxidant activity seemed negligible, and no cytotoxicity arose. Conclusions: PLL-EO exhibits a broad-spectrum activity against periodontal bacteria and *Candida*, with an interesting dual inhibitory capacity toward COX-2 and LOX inflammatory enzymes, and without side effects against oral cells.

## 1. Introduction

Periodontal and peri-implant diseases, in addition to *Candida* spp. infections, are among the most common and recalcitrant oral plights that evoke immuno-inflammatory responses. Evidence is provided that periodontitis is associated with an imbalance of the oral microbiome [1] and other risk factors such as smoking, hyperglycaemia, and xerostomia [2,3]. Within the oral microbiome imbalance, Gram-negative, proteolytic, and anaerobic biofilm-forming bacteria, namely *Porphyromonas gingivalis*, *Tannerella forsythia*, and *Fusobacterium nucleatum*, may become dominant, settle gingival margins, and pave the way for chronic multifactorial disease [1,4,5], whereas Candidiasis by *Candida albicans* and *Candida glabrata* turns out to be among the predominant infective-agents in immunocompromised patients [6]. Following epithelial cell harming by pathogenic agents, the inflammatory process sets off, and pro-inflammatory cytokines are released to start the healing process [1,7,8]. However, under these physio-pathological conditions, severe and recurrent inflammation can be concomitant with periodontitis and oral Candidiasis [9,10], and may contribute to deleterious consequences. Lasting oxidation of arachidonic acid by cyclooxygenases (COX-1 and COX-2) and lipoxygenase (5-LOX) leads to the accumulation of prostaglandins and leukotrienes [11,12]. Among these oxidized products, prostaglandine E_2_ (PGE_2_) and related lipid mediators activate immune cells and stimulate pro-inflammatory and pro-coagulant responses. Furthermore, the activation of phagocytes within the inflammation process burst oxygen consumption, jeopardizing severely cell redox homeostasis by rising reactive oxygen species (ROS) [13]. Within this framework, the repair of the infected tissue is hindered, favouring the outbreak of chronic multifactorial diseases [14,15,16,17]. 

With the intent to antagonise recurrent oral inflammation, oral antiseptics are often administered. However, several synthetic molecules, such as chlorhexidine digluconate, largely used in commercial oral antiseptics, have shown high cytotoxicity against human fibroblasts and osteoblasts [18,19,20,21]. Time-dependent toxicity was demonstrated in fibroblasts viability after exposure to chlorhexidine >0.001% [19], and cellular damages were raised by the presence of other chemicals in the mixture attesting an increase in toxicity by synthetic substances [21]. On the other hand, if employing systemic non-steroidal anti-inflammatory drugs (NSAIDs) or even steroids, gastrointestinal and cardiac toxicity as well as nephrotoxic side effects can occur [22,23,24]. 

Thus, new pharmacological molecules differently acting, with a broad-microbial targeting activity, and able to constraint also the inflammation process without producing side effects to the host, are needed [25]. In this regard, plant essential oils (EO) are potential candidates, even though, according to concentration, toxicity towards cells has been reported [13,26]. Essential oils are phytocomplexes composed of several chemical classes, among which polyphenols are acquiring an increasing interest due to their functional properties [27,28,29,30,31,32]. Concerning these phytocomplexes, *Pistacia lentiscus* L. (PLL), belonging to the Anacardiaceae family, is a wild-growing species of the Mediterranean basin particularly diffused in Sardinia (Italy), with leaf-EO rich in terpenoids [33]. In popular medicine, PLL plant and processed products (leaf-EO, drupe oil, resin) have been largely employed as oral antiseptic, anti-inflammatory, analgesic, and healing agents [33,34]. Quartu et al. (2012) [35] reported that PPL-leaf-EO promoted the biosynthesis of highly polyunsaturated fatty acids in mice, lowered the expression of COX-2, and protected brain tissue from ischemia, while Orrù et al. (2017) [36], employing PPL-drupe oil, evidenced a selective growth inhibition towards some pathogenic bacteria of the oral cavity. 

Given the above considerations, after determining the major constituents of PLL leaf essential oil (PLL-EO), the purpose of this study was to evaluate its capacity for some functional implications regarding the periodontal disease and *Candidiasis*. With this intention, we assessed the inhibitory ability to PLL-EO to a wide range of microorganisms and clinical species involved in these infections, in addition to its anti-inflammatory strength. Further, we tested potential cytotoxicity of PLL-EO against the periodontal ligament fibroblasts (PDLF), gingival fibroblasts (GF), gingival keratinocytes (GK), and dysplastic oral keratinocytes (DOK). 

The hypothesis of the present study was that PLL-EO has anti-microbial and anti-inflammatory activities without negatively interfering with human cells viability.

## 2. Results

### 2.1. Essential Oil Yield and Chemical Fingerprinting

Following hydro-distillation, the yield of PLL-EO was about 0.41% (w/w). The GC-MS analyses identified 64 constituents, which represented 97.28% of the total components (Table 1). Among these, 17 components made up 1% of the total. GC-MS identified α-pinene and terpinen-4-ol as the main constituents of PLL-EO with a concentration of 16.89% and 16.49%, respectively. The monoterpenes, limonene and β-myrcene, represented 3.89% and 0.87%, respectively, while the sesquiterpenes, (Z)-caryophyllene, and (E)-caryophyllene, embodied 1.39% and 0.07%, respectively.

### 2.2. Minimal Inhibitory Concentrations Against Oral Bacteria and Candida sp.

The MIC of PLL-EO (Table 2) against oral bacteria involved in periodontal diseases ranged between 3.13 µg/mL (*P. gingivalis*, *T. forsythia*) and 12.5 µg/mL (*S. gordonii*). Regarding the effects on *Candida* sp., the recorded values ranged from 6.25 µg/mL for *C. glabrata* strains to 12.5 µg/mL for *C. albicans* ones.

### 2.3. Anti-inflammatory Assays

#### 2.3.1. COX-1/2 Inhibition

The anti-inflammatory activity, expressed as a percentage of COX-1/2 inhibition by increasing concentrations of PLL-EO (1, 10, and 100 μg/mL), allowed us to establish the IC_50_ values. Compared to the control sample (0 PLL-EO), where the COX-1/2 activity was considered equal to 100% (Figure 1), the IC_50_ values resulted as 10.3 ± 4.4 μg/mL and 6.1 ± 2.5 μg/mL for COX-1 and COX-2, respectively. These values were only seven times higher than the IC_50_ values of the positive control ibuprofen (1.3 ± 0.5 μg/mL for COX-1, 0.87 ± 0.39 μg/mL for COX-2). Thus, the inhibitory activity of PLL-EO toward COX-2 was higher in comparison to that produced toward COX-1, similarly as it was regarding ibuprofen.

#### 2.3.2. LOX Inhibition

The measured values were related to a control sample, which represented 100% enzyme activity, i.e., uninhibited reaction. According to the data (Figure 2), the highest PLL-EO concentration employed (100 μg/mL) did not reach the IC_50_ value since the activity was lowered by 30% compared to the control. Thus, PLL-EO showed a weak inhibitory effect toward LOX activity. Phenidone (100 μg/mL) used as positive control inhibited LOX activity by 63.4 ± 8.6%.

#### 2.3.3. Electro-Chemical Determination of the Antioxidant Activity

According to previous studies, the redox potential of +500 mV is used to detect the antioxidant capacity of phenolic compounds in vegetable matrices, while +800 mV potential is used to assess the total phenolics content [37,38]. A low oxidation potential indicates a high reducing power since the ionization potential is the main factor to determine the efficiency of antioxidants [39]. In Figure 3, the voltammogram of PLL-EO was compared with that of *α*-tocopherol (green line), a well-known antioxidant molecule. The red line indicates PLL-EO oxidation occurred at +550 mV, where it splits the baseline (black line) with an oxidation peak of about +800 mV. Thus, the data suggests a poor antioxidant activity of the hydro-distilled EO. Cyclic voltammetries of PLL-EO main components further accredit this result: *α*-pinene and terpinen-4-ol split the baseline at +600 mV and +450 mV, with peaks at +720 mV and +750 mV respectively, while no redox peaks were observed regarding *α*-phellandrene (Appendix A).

### 2.4. Viability Assay

The protective effects of PLL EO toward the oxidative stress affecting the GF were evaluated by the MTT assay, testing different concentrations of the oil. The results showed a significant increase in cell viability when GFs were treated with 10, 25, 50 and 75 µg/mL of PLL EO in comparison to that observed in the control group and using the DMSO 0.1%. Differently, the cells subjected to the oxidative stress evidenced a vitality drop of about 30% in comparison to the control, and when PLL EO was administered before the oxidative stress, no beneficial effects occurred (Figure 4).

### 2.5. Cytotoxicity Assay

Cytotoxicity effects of increasing concentrations of PLL-EO were recorded for PDLF, GF, GK, and DOK cell lines following a 24 h incubation. The data did not evidence significant differences (*p* < 0.05) between PLL-EO treated cells and the control. Thus, according to the highest concentration employed in this experiment, it is possible to claim that no toxicity in oral cell lines occurred up to 100 μg/mL of the oil (Figure 5).

## 3. Discussion

New treatment options for infections and inflammation should target selectively the pathogens, preventing their ability to form biofilms [40]. Further, they should modulate the host inflammatory response in order to antagonize the accumulation of PGE_2_ and leukotrienes in chronic inflammatory diseases [11,41]. Moreover, there is a demand for biocompatibility to prevent side effects on the host [21,25]. Natural products are a valuable source of biomolecules that in view of their chemical versatility, offer the possibility to synthesize new drugs containing all of the above-mentioned capacities [14,27,28,29,30,31,32].

In the present study, we evaluated the peculiar character of PLL essential oil derived from the leaves of wild plants growing in the North of Sardinia. The data demonstrated PLL-EO possesses antimicrobial activity against the tested species in addition to anti-inflammatory properties and high biocompatibility. The antimicrobial capacity was studied by the broth microdilution method, according to the recommendations of EUCAST, in order to compare the data to published reports. Similarly, an enzymatic cell-free in vitro assay using human recombinant COX enzymes and using soybean LOX was carried out to evaluate the inhibitory activity against the most relevant inflammatory mediators [27]. The antioxidant activity was assessed using an electron-chemical method measuring this capacity as the redox properties of the raw oil and the most represented compounds. The electron-chemical system has been credited as a very good device in such a field thanks to the high sensitivity, chemical inertness, and versatility [37,38,39,42]. In addition, the capacity of selectivity measures the anti-oxidant potency of different fractions in a complex makes the electro-chemical an innovative benchmark method. This is an advantage in comparison to the spectrometric DPPH (2,2-diphenyl-1-picrylhydrazyl) stable assay, which measures the anti-redox potential of the whole pool of compounds in an agent [43]. In addition, the anti ROS effect of the oil was evaluated using GF and the biological MTT method, which is widely applied to ascertain the capability of plant extracts to protect cells of different origins toward H_2_O_2_ stress [28,44]. Finally, the biocompatibility was assessed by means of WST-1 metabolic assay using four oral cell type lines. The assay allowed us to study the metabolic activity of each cell line after contact with increasing concentrations of PLL-EO, as the viability of mammalian cells grown in culture is largely applied as an indicator of potential toxic effects in animals [44]. In addition, considering that any substance, particularly oral antimicrobials, may potentially come into direct contact with a wide range of cells, we extensively conducted the safety assessment of PLL-EO by testing PDLF, GF, GK, and DOK cells.

PLL is an evergreen environmentally-sustainable shrub, well adapted to harsh growing conditions, dryness, and a warm environment, which all exercise an influence on the genotype [33]. PLL is rich in alternate, leathery leaves, which represent the easiest means to extract the essential oil on a large scale. Thus, the choice to use PLL-EO was aimed to produce highly-repeatable results in view of further researches pointing to novel therapeutic formulations of the oil. 

As it has been shown by the GC-MS, the PLL-EO has a high content of terpenes. The most significant fractions were *α*-pinene and terpinen-4-ol constituting 33.38% of the total EO; furthermore, limonene (3.89%) and *β*-myrcene (0.87%); (Z)-caryophyllene (1.39%) and (E)-caryophyllene (0.07%) were detected (Table 1). These natural compounds are a matter of increasing interest in investigations [31,32] and were recovered in the raw EO in a concentration above 0.05%, which allows us to classify PLL-EO as pharmacologically-active [31]. The chemical composition of PLL-EO depends on the geographical region. Our PLL-EO was high in α-pinene and sabinene content when comparing with PLL-EO collected in Turkey [45] or even in other regions of Sardinia [35]. Both α-pinene and sabinene were described to exert strong antibacterial activity [46,47], and additionally, α-pinene protects cells against oxidative stress [48]. Comparing the chemical profile of the raw EO with that was reported in a previous analysis regarding PLL leaves-EO from Sardinian biodiversity [33], it can be noted that, over time, and regardless of harvesting and seasons, the concentration of such terpenoids has never been lower than 0.05%. Consequently, our data attest to a fairly constant chemical profile of PLL leaves-EO. 

The EO exerted antimicrobial activity against all the tested microorganisms (Table 2). It is interesting to note that its activity was higher when compared to that reported in regard to isolated fractions of the oil derived from plants growing in other Mediterranean areas [49,50]. This result can suggest a synergy existing between the pharmacological fractions of terpenes, which characterize the chemotype of PLL-EO we have assessed. High antibacterial activity was found against the reference strains and clinical isolates of *P. gingivalis* and *T. forsythia* (MICs 1.63–3.13 µg/mL). Moreover, the oil exerted activity toward other important species, e.g., *F. nucleatum*, which is a co-aggregating bacterium in oral biofilms [51], further releasing bone-resorbing pro-inflammatory cytokines and chemokines [10], and to a less extent, toward the commensal *S. gordonii*. This antimicrobial activity would allow the possible use of PLL to antagonize gingivitis as a primary strategy to prevent periodontitis and a secondary preventive strategy to recurrent periodontitis after periodontal surgery [3]. 

The possibility to formulate PLL-EO as a potential therapeutic or antiseptic is attractive as testing the viability of GF by the MTT assay, it increased in the presence of the EO in comparison to that observed in the control and DMSO (Figure 4). This evidence can indicate a possible revitalizing ability of the oil toward oral fibroblasts. Our findings are emphasized by the fact that WST-1 did not demonstrate cytotoxicity toward any of the evaluated cell cultures (Figure 5). A safe behavior of PDLF moreover strengthens the use of PLL-EO as an antimicrobial in periodontal disease without interferences in the immune and reparative events, in which fibroblasts are actively involved during the infection [1,10]. Additionally, the biocompatibility with PDLF and GF may put extra emphasis on the capacity of the oil in wound healing because these fibroblasts, as mesenchymal cells, perform important vital functions and are responsible for much of the synthesis of new extracellular matrix during the reparative procedures [1,10]. In regard to the biocompatibility with GK and DOK, it highlights PLL as an antimicrobial with respect for stem cell differentiation and any stimulus toward dysplastic or pre-malignant oral mucosa cells. All these results are appealing, especially because more and more reports have evidenced that synthetic antimicrobials in several oral antiseptics possess cytotoxic effects and suppress the viability of oral cells [19,20,21]. 

We further demonstrated that the PLL-EO was capable of lowering the activity of COX-2 and LOX enzymes (Figure 1 and Figure 2), which are specifically produced in the course of an active inflammatory process [11,12], like that which is developed during the periodontal infection [1,10]. Our in vitro data is supported by in vivo studies that reported PLL-EO reduced the expression of COX-2 in experimental rats, thus preventing harmful effects connected to eicosanoids accumulation in injury [35].

While the study reported here can suggest that PLL-EO could be considered as a useful phytocomplex to treat causes and effects characterizing the periodontal disease, and in regard for oral cells biocompatibility, the antioxidant activity which resulted was low (Figure 3 and Figure 4). This data seems not to be in accordance with previous reports [33,35,44], which measured the anti-ROS capacity of PLL growing in Sardinia using the DPPH spectrophotometric assay and tested the antioxidant protection of different types of PLL extracts on different cell lines in comparison to those used in the present research. Chemically, disagreement can be explained through the different principles that regulate the measurement of the total quantity of antioxidant molecules: the redox properties of phenolics adopted by the electro-chemical method, and the ability of phenolics to scavenge the radical by the DPPH assay [38]. In addition, the simultaneous measurement of the anti-redox potency of the whole pool of compounds in an agent by the DPPH can lead to overestimation of the antioxidant activity of individual chemical classes [43]. With regard to the antioxidant protection of PLL-EO toward H_2_O_2_ damages in cell cultures, the replay could be different in in vivo experimentations and under different testing conditions [44]. This assumption is supported by the fact that PLL-EO was capable of reducing oxidation in frontal cortex cells of adult Wistar rats submitted to carotid artery obstruction [35]. Thus, this aspect should be further defined in future in vivo and in silico assessments. 

Furthermore, PLL-EO evidenced antimicrobial activity against reference strains and clinical isolates of *C. albicans* and *C. glabrata* (Table 2). The MIC values of PLL-EO were low against *Candida* sp., particularly in the case of *C. glabrata*, which is known for its clinical resistance to antimicrobials [52]. In fact, *Candida* tends to withstand anti-fungal treatments, especially when biofilms are formed. This is probably due to the ability of the yeast to produce a high quantity of fungal PGE_2_, which also exhibits cross-reactivity with that of the host [9]. The issues are increased because of the side effects of antifungal therapies in humans, which may occur due to the poor ability of the drugs to target selectively the fungal cell [9]. In an attempt to solve the problems connected to the yeast resistance, it was demonstrated that the use of aspirin and other COX inhibitors (i.e., NADPHs), increased the anti-fungal efficacy of some compounds, particularly against biofilm cells [53]. Results suggest that COX-2 inhibitors are able to antagonize the biofilm development, resistance, and invasion of the yeast by lowering the level of PGE_2_. In a similar way, LOX inhibitors showed activity toward fungal PGE_2_ in a dose-dependent manner [9]. The finding reported here about PLL-EO inhibitory activity toward COX-2 and LOX may be indicative of its capacity to regulate the aggressiveness of *Candida* sp. Thus, even if further studies are needed to evaluate the specific activity of the EO towards biofilm development, the low MIC values, in addition to the evidenced impact on the most important inflammatory mediators, should indicate PLL-EO as a good antifungal agent, which acts directly against the yeast and indirectly against its virulence with no oral cytotoxicity. This point will be a matter of our future researches.

## 4. Materials and Methods

### 4.1. Chemicals and Abbreviations

Thiazolyl blue tetrazolium bromide (MTT); isopropanol; streptomycin/penicillin; hydrogen peroxide (H_2_O_2_); gentamicin; phosphate-buffered saline (PBS) were obtained from Sigma-Aldrich (Milan, Italy). Dulbecco’s modified Eagle’s medium (DMEM) and Fetal Bovine Serum (FBS) were purchased from Euroclone S.p.A. (Pero, Milan); dimethyl sulfoxide (DMSO).

### 4.2. Plant Material, Essential Oil Extraction, and GC/MS Characterization

In October 2017, Pistacia lentiscus leaves (PLL) were harvested from wild plants growing on an acidic-siliceous soil under typical Mediterranean climate conditions (Costa Paradiso-Sardinia, 41°−3′−21″ N, 8°−57′−14″ E). October was chosen because harvesting at this time of the year allows attaining the highest amount of young-mature metabolic active leaves, before the winter senescence takes place. The collected leaves were immediately cold stored (8 °C), transported to the laboratory, and processed within 24 h. Before extraction, 3-uniform replicates of the leaves were prepared. Then, a 4 h-hydro-distillation took place by a Clevenger-type apparatus, as the method of chose to obtain the EO [54]. Following water removal, the recovered oil fraction termed ‘essential oil’ (PLL-EO) was stored at −20 °C until analyses. 

### 4.3. Gas Chromatography -Mass Spectrometry (GC-MS) Analysis

Replicates of the PLL-EO were separately analyzed using a GC (Hewlett-Packard Model 5890A) equipped with a flame ionization detector and fitted with a 60 m × 0.25 mm, thickness 0.25 mm ZB-5 fused silica capillary column (Phenomenex). The injection port and detector temperature were 280 °C. The column temperature was subjected to a linear rise (5 °C/min) starting from 50 °C with 3 halts: 1 min at 135 °C; 5 min at 225 °C and a final 10 min halt at 260 °C. Oil samples of 0.2 μL (injection volume) were analyzed, diluted in hexane using 2,6-dimethylphenol as an internal standard. Injection was performed using a split/splitless automatic injector HP 7673 and helium as a carrier gas. Several measurements of peak areas were performed through a HP workstation with a threshold set to 0.00 and peak width to 0.02. Compound quantification was expressed as absolute weight percentage using internal standard response factors (RFs). Since oxygenated compounds have lower detectability than hydrocarbons by the Flame Ionization Detector (FID), detector RFs were determined for key components relative to 2,6-dimethylphenol and assigned to other components on the basis of the functional group and/or structural similarity. For this reason, we have multiplied the obtained data for the following correction factors: hydrocarbons by 1; aldehydes and ketones by 1.24; alcohols by 1.28 and esters by 1.408.

GC-MS analysis was carried out with an Agilent Technologies model 7820A connected to a MS detector 5977E MSD (Agilent) using the same column and operative conditions described formerly. The column was connected with the ion source of the mass spectrometer. Mass units were monitored from 10 to 900 AMU at 70 eV. The identification procedure was carried out exclusively considering peaks ranging from 40 to 900 AMU. Compound identification was based on Rt values and mass spectra comparison with those obtained from authentic samples and/or NIST and Wiley library spectra, or the interpretation of EI-fragmentation of molecules [55]. 

### 4.4. Minimal Inhibitory Concentration Assay

According to the recommendations of EUCAST, the microbroth dilution technique was used to measure the minimal inhibitory concentration. The inhibitory activity of PPL-EO was tested against *Streptococcus gordonii* (ATCC 10558); *Actinomyces naeslundii* (ATCC 12104); *F. nucleatum* (ATCC 25586); *P. gingivalis* (ATCC 33277 and two clinical isolates); *T. forsythia* (ATCC 43330 and two clinical isolates); *C. albicans* (ATCC 76615 and two clinical isolates) and *C. glabrata* (DSM 6425 and two clinical isolates). In short, PLL-EO was added in a two-fold dilution series beginning from 200 µg/mL. As culture media, Wilkins-Chalgren broth was employed for all bacterial strains except for *S. gordonii* cultured in an adjusted Mueller-Hinton broth. For *Candida* sp. the RPMI-medium was used. According to EUCAST, after 18 h of incubation at optimal growing conditions, MIC was determined visually as the lowest effective concentration of PLL-EO without microbial development. Experiments were made in independent replicates (n = 3). A growth control containing distilled water instead of PLL-EO was also included.

### 4.5. Anti-Inflammatory Assays

#### 4.5.1. COX-1/2 Inhibition

To determine the anti-inflammatory activity of 1.0, 10.0, or 100 μg/mL PLL-EO, an enzymatic cell-free in vitro assay using human recombinant cyclooxygenase (COX-1, COX-2) (Sigma-Aldrich; Prague, Czech Republic) was performed [27]. The activity was expressed as the percentage of cyclooxygenase inhibition by PLL-EO compared to untreated samples (0.0 μg/mL PLL-EO). Briefly, COX-1 (1 unit/reaction) or COX-2 (0.2 unit/reaction) was added to 180 μL of the incubation mixture (100 mM Tris buffer pH 8.0; 5 μM porcine hematin; 18 mM L-epinephrine; 50 μM Na_2_EDTA) in a 96-well plate. PLL-EO (1.0, 10.0, and 100 μg/mL) was dissolved in ethanol (10 μL) and the reaction was started with 10 μM arachidonic acid. Ibuprofen (1.0, 10.0, or 100 μg/mL) was used as the positive control. The reaction was stopped after 20 min by adding formic acid (10%). The main product of this reaction, PGE_2_, was quantified using a PGE_2_ ELISA kit according to the manufacturer’s instructions (Enzo Life Sciences, New York, NY, USA). The absorbance was recorded at 405 nm (Tecan Infinite M200 microplate reader). Experiments were repeated three times with three technical replicates.

#### 4.5.2. LOX Inhibition

An enzymatic cell-free in vitro assay using soybean LOX (Sigma-Aldrich; Prague, Czech Republic) was carried out to test the inhibitory activity of 1.0, 10.0, 50.0, and 100 μg/mL PLL-EO [27]. The activity was expressed as the percentage of inhibition compared to the untreated samples (0.0 μg/mL PLL-EO). Briefly, the reaction mixture in a 96-well plate was made of LOX (0.8 units); 0.2 M borate buffer pH 8.5 (190 μL); PLL-EO (1.0–100 μg/mL) dissolved in ethanol, and linoleic acid (90 μM) was added as enzyme substrate to start the reaction. Phenidone (1.0, 10.0, 50.0 and 100 μg/mL) was used as the positive control. The inhibitory activity of PLL-EO on LOX was assessed by recording the absorbance of the conjugated diene formed from linoleic acid at 234 nm (Tecan Infinite M200 microplate reader). 

#### 4.5.3. Antioxidant Activity

The electro-chemical characterization of PLL-EO was performed according to Barberis et al. 2010 [42] with some modifications, using a four-channel system (eDaQ Quadstat, e-Corder 410 and Echem software, eDAQ Europe Poland) and screen-printed sensors (Metrohm Italiana s.r.l.) consisting of a 4 mm working electrode (WE), an Ag/AgCl reference electrode (Ref) and a carbon counter electrode. Air-bubbled PBS at pH = 7.4 was used as a supporting electrolyte. Voltammograms were obtained placing a drop of 70 μL solution on the sensor surface. Cyclic voltammetries (CV), carried out in order to investigate the PLL EO electro-chemical behavior at the WE surface vs. the Ag/AgCl Ref, were executed starting from −200 mV to +800 mV, at a scan rate of 100 mV, in the absence (PBS + 1% DMSO) and in the presence of PLL-EO and α-tocopherol, as the reference antioxidant compound. In order to have a 200 mg/mL oil concentration, a solution of PLL-EO was prepared in PBS containing 1% DMSO. In addition, CVs of α-pinene, terpinen-4-ol and α-phellandrene standards (Sigma-Aldrich, Milan, Italy), as the three main represented compounds in the EO, were carried out using the same methodology starting from −1 V to +1 V. 

### 4.6. Viability Assay 

#### 4.6.1. Cell origin, Culture, and Treatments

The research was ethically conducted in accordance with the Declaration of Helsinki. The protocol and informed consent forms were approved by the Ethics Committee at the University of Sassari [n° 1000/CE] [56]. Gingival tissues were attained from donors subjected to wisdom teeth surgery at the Department of Surgery of the Dental Unite, University of Sassari, Italy. Donors were healthy subjects, not suffering from periodontitis or other chronic diseases. Patients were carefully informed of the study’s purpose, risks, and benefits. Informed consent was obtained from all subjects prior to the study. 

Gingival primary fibroblasts were isolated from the above-mentioned gingival tissue. GF were cultured in Dulbecco’s modified Eagle’s medium containing 10% FBS and 1% penicillin (100 U)/streptomycin (100 μg/mL) at 37 °C under humidified 5% CO_2_/air. Then, 2 × 10^3^ GF/100 µL (2nd–4th passage) were plated in 96-well plates and amended with increasing concentrations of PLL-EO (0.0, 10, 25, 50, 75 and 100 μg/mL) dissolved in 0.1% DMSO. Immediately, plates were returned to the incubator for 24 h. At this time, 100 µM H_2_O_2_ was added and the viability was determined by the MTT assay after 140 min. The post-oxidative stress recovery effect of PLL-EO was checked using the same experimental plan with the exception that GF cells were kept first amended with 100 µM H_2_O_2_ for 24 h, and then treated with PLL-EO at the given concentrations before performing the MTT assay after 140 min.

#### 4.6.2. MTT Assay 

GF viability was evaluated by the MTT reduction assay [57]. In short, GF cells were incubated with 100 µL (0.05 mg) of MTT for 3 h at 37 °C. Then, MTT was removed, and the precipitated blue formazan crystals were dissolved in 100 µL of isopropanol and the color was read at 570 nm using a microplate reader (EMax^®^ Plus, Molecular Devices). Cell growth was calculated by normalizing the absorbance of treated cells to the corresponding control. All the experiments were performed in quadruplicate and repeated at least three times. 

### 4.7. Cytotoxicity Assay

The WST-1 metabolic activity assay was conducted using PDLF (ScienCell Research Laboratories, Carlsbad, CA, USA, Cat. No. 2620), GF (ScienCell, Cat. No. 2630), GK (The European Collection of Authenticated Cell Cultures – ECACC, Cat. No. PCS-200-014), and DOK (ECACC, Cat. No. 94122104) cells. All cells were seeded in 96-well plates (6 × 10^3^ cells/well) employing DMEM with 10% heat-inactivated FBS. After 24 h, the culture medium was replaced by the same medium amended by increasing concentrations of PLL-EO (1.0, 10, 25, 50, 75, and 100 µg/mL) dissolved in DMSO (0.1%) and ethanol. Control wells were supplemented by the sole medium with 0.1% DMSO and ethanol. After 24 h of exposure, the medium was removed, and the cells were rinsed twice with 150 μL PBS. Then, according to the manufacturer’s instructions, 100 μL of culture medium without FBS containing 0.3 mg/mL cell proliferation reagent WST-1 (Roche, Sigma-Aldrich) was added to each well. Absorbance was measured immediately at 450/650 nm (Tecan Infinite M200 microplate reader) and repeated after 2 h. Metabolic activity of each cell line was assayed in six parallels and in three independent experiments. Viability of the treated cells was expressed as a percentage referred to untreated controls (viability 100%). The calculations were done using GraphPad Prism 8.1.0 (GraphPad Software, San Diego, CA, USA).

## 5. Conclusions

Herein, we documented the antimicrobial capacity of PLL-EO, which exhibits a broad-spectrum activity against periodontal bacteria and *Candida* as reference strains and *clinical* sp., antagonizing them with low MICs. Furthermore, our in vitro study reported that PLL-EO has inhibitory activity toward arachidonic acid oxidative enzymes, specifically proposing the oil as an interesting dual inhibitory agent against COX-2 and LOX. Data suggest a possible role of the EO in the resolution of the inflammatory process, which accompanies periodontal and *Candida* infections. These facts also sustain that PLL-EO could be a novel antifungal agent that acts directly against the yeast and indirectly against its virulence. All the above-mentioned considerations, further supported by the wide biocompatibility in oral cells, validate our hypothesis and underline the potential of PLL-EO obtained from leaves collected at Costa Paradiso-Sardinia, as an ingredient in oral health-care products for the long-term use. Such oral-health care products may have beneficial effects in periodontal therapy, as they may prevent gingival inflammation and Candida infections, in particular in immune-compromised patients.

## Figures and Tables

**Figure 1 antibiotics-09-00281-f001:**
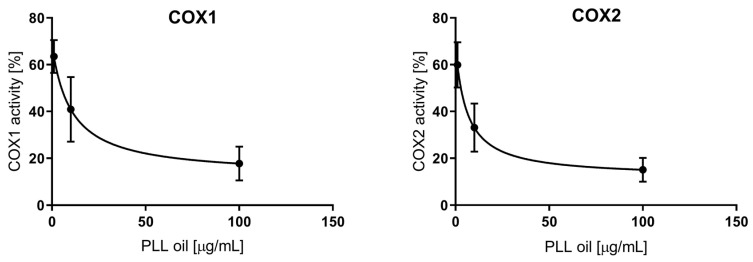
The anti-inflammatory effect of PLL-EO toward cyclooxygenase (COX-1 and COX-2) activity.

**Figure 2 antibiotics-09-00281-f002:**
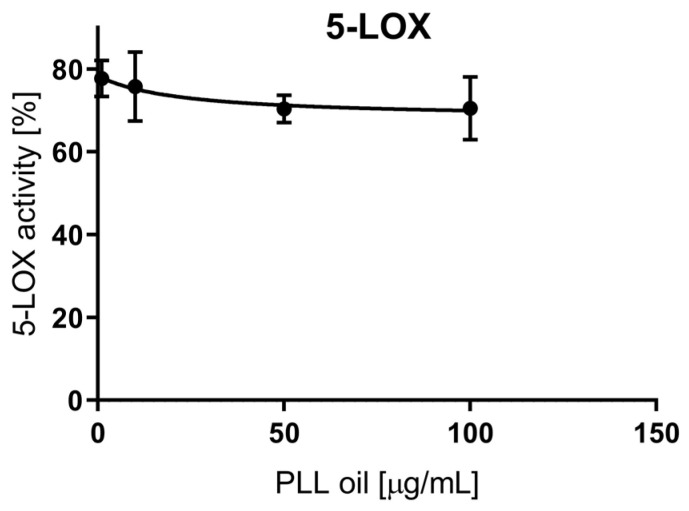
The anti-inflammatory effect of PLL-EO toward soybean lipoxygenase (LOX) activity.

**Figure 3 antibiotics-09-00281-f003:**
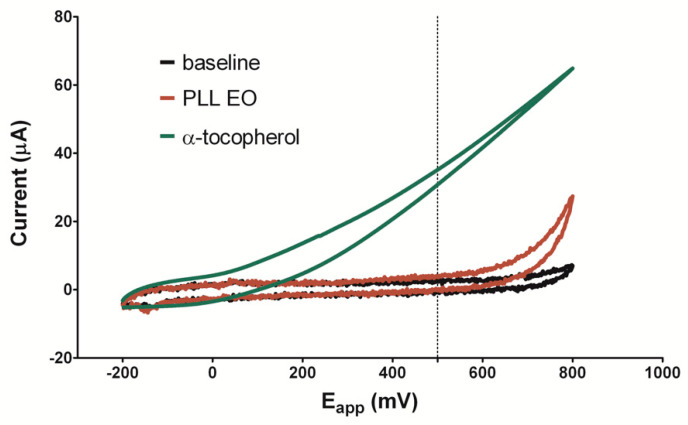
Cyclic voltammetry of PLL-EO with a scanned potential range (E_app_) between −200 mV and +800 mV *vs* Ag/AgCl reference electrode, in the absence (black line) and in the presence of 200 mg/mL PLL-EO (red line), in comparison with 1 mM α-tocopherol (green line).

**Figure 4 antibiotics-09-00281-f004:**
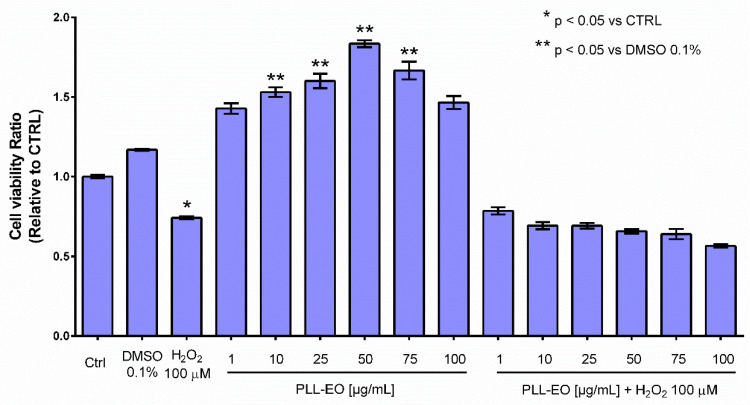
Primary gingival fibroblasts viability relative to the sole growth medium (CTRL), if supplemented with 0.1% DMSO or 100 µM H_2_O_2_ and when amended with PLL-EO − H_2_O_2_ and PLL-EO + H_2_O_2_. The separation of mean values (n = 4) and standard deviation were calculated according to the Student-Newman-Keuls test at *p* ≤ 0.05.

**Figure 5 antibiotics-09-00281-f005:**
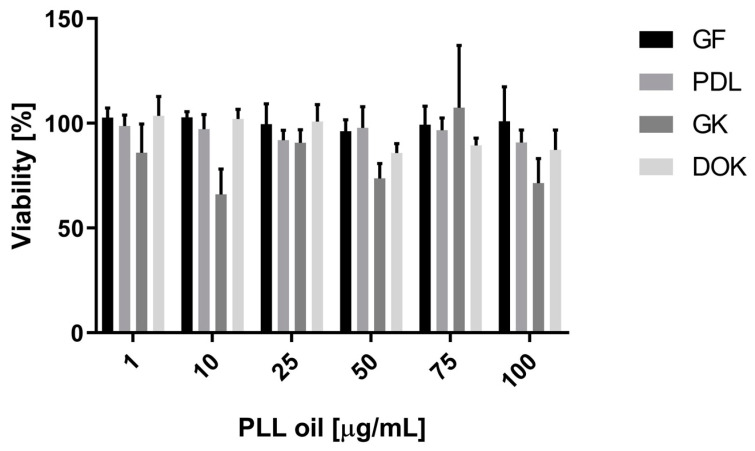
The effect of increasing concentrations of PLL-EO on the viability of human oral cells following 24 h incubation. GF: gingival fibroblasts; PDLF: periodontal ligament fibroblasts; GK: gingival keratinocytes, and DOK: dysplastic oral keratinocytes. Data (mean from six parallels in three independent experiments ± S.D.) are expressed as the percentage of viability of untreated cells (=100%).

**Table 1 antibiotics-09-00281-t001:** Chemical composition of *Pistacia lentiscus* L. essential oil from fresh leaves.

RI apol Sper	RI apol Lett	Constituents	%	ID *
924	927	tricyclene	0.24 ± 0.01	MS-RI
930	931	*α*-thujene	0.23 ± 0.01	Std
939	937	*α*-pinene	16.89 ± 0.15	Std
955	956	camphene	1.04 ± 0.04	Std
977	975	sabinene	7.73 ± 0.11	Std
981	979	*β*-pinene	4.30 ± 0.05	Std
992	991	*β*-myrcene	0.87 ± 0.02	Std
1005	1003	*α*-phellandrene	7.39 ± 0.12	Std
1015	1017	*α*-terpinene	4.79 ± 0.04	Std
1025	1025	p-cymene	1.10 ± 0.03	Std
1027	1029	limonene	3.89 ± 0.07	Std
1031	1030	*β*-phellandrene	4.71 ± 0.05	MS-RI
1035	1037	*Cis**-β*-ocimene	0.24 ± 0.02	MS-RI
1056	1056	isoamyl isobutyrate	0.55 ± 0.02	MS-RI
1056	1056	2-methylbuthyl butanoate	0.18 ± 0.01	Std
1064	1060	*γ*-terpinene	6.30 ± 0.09	Std
1087	1089	terpinolene	3.25 ± 0.02	MS-RI
1090	1091	p-cymenene	0.12 ± 0.01	MS-RI
1094	1094	isopentyl isivalerate	0.13 ± 0.01	MS-RI
1108	1108	n-amyl isovalerate	0.04 ± 0.01	MS-RI
1115	1117	fenchol	0.11 ± 0.01	MS-RI
1147	1144	*Cis*-*β*-terpineol	0.13 ± 0.01	Std
1166	1169	borneol-endo	0.16 ± 0.01	Std
1181	1177	terpinen-4-ol	16.49 ± 0.18	MS-RI
1187	1189	*α*-terpineol	3.98 ± 0.07	MS-RI
1194	1199	*γ*-terpineol	0.07 ± 0.01	MS-RI
1242	1238	isopentyl hexanoate	0.24 ± 0.02	MS-RI
1247	1247	2-methylbuthyl hexanoate	0.17 ± 0.01	Std
1258	1253	piperitone	0.05 ± 0.01	Std
1294	1294	2-undecanone	0.85 ± 0.03	MS-RI
1355	1351	*α*-cubebene	0.05 ± 0.01	Std
1378	1377	*α*-copaene	0.24 ± 0.01	MS-RI
1380	1382	*β*-maaliene	0.33 ± 0.02	MS-RI
1407	1409	(Z)-caryophyllene	1.39 ± 0.06	MS-RI
1419	1419	(E)-caryophyllene	0.07 ± 0.01	Std
1457	1460	alloaromadendrene	0.14 ± 0.01	Std
1480	1480	*γ*-muurolene	0.57 ± 0.03	Std
1485	1485	*α*-amorfene	0.07 ± 0.01	MS-RI
1486	1485	germacrene D	2.73 ± 0.11	MS-RI
1497	1499	*Cis*-dihydro apofarnesal	0.14 ± 0.01	MS-RI
1502	1500	*α*-muurolene	0.53 ± 0.02	Std
1509	1506	*β*-bisabolene	0.19 ± 0.01	MS-RI
1521	1523	*δ*-cadinene	1.43 ± 0.04	MS-RI
1535	1535	*Trans*-cadina-1(2),4-diene	0.15 ± 0.01	Std
1539	1539	*α*-cadinene	0.07 ± 0.01	Std
1552	1552	*Cis*-muurol-5-en-4-*β*-ol	0.07 ± 0.01	Std
1535	1535	*Trans*-cadin-4-en-7-ol	0.31 ± 0.02	Std
1639	1640	epi-*α*-cadinol	1.22 ± 0.08	Std
1644	1646	*α*-muurolol	0.94 ± 0.04	Std
1648	1652	cedr-8(15)-en-9-*α*-ol	0.12 ± 0.01	Std
			96.96	

***** ID = Identification methods: MS by comparison of the mass spectrum with those of the computer mass libraries Adams (NIST 11) and by interpretation of the mass spectra fragmentations; RI by comparison of retention index with those reported in literature; Std by comparison of the retention time and mass spectrum of available authentic standards; MS identification of mass spectrum. No-polar column ZB-5. Data are the mean of three replicates.

**Table 2 antibiotics-09-00281-t002:** MIC (µg/mL) of *Pistacia lenticus* L. essential oil against oral microbiota.

Strain	Origin	MIC (µg/mL PLL-EO)
*Streptococcus gordonii* ATCC 10558	Laboratory	12.5
*Actinomyces naeslundii* ATCC 12104	Laboratory	3.13
*Fusobacterium nucleatum* ATCC 25586	Laboratory	6.25
*Porphyromonas gingivalis* ATCC 33277	Laboratory	3.13
*P. gingivalis* BeOR6	Clinical isolate	1.63
*P. gingivalis* BeOR14	Clinical isolate	1.63
*Tannerella forsythia ATCC 43330*	Laboratory	3.13
*T. forsythia* Be13237	Clinical isolate	1.63
*T. forsythia* Be13216	Clinical isolate	3.13
*Candida albicans* ATCC 76615	Laboratory	12.5
*C. albicans* BeT41	Clinical isolate	12.5
*C. albicans* BeT603	Clinical isolate	12.5
*Candida glabrata* DSM 6425	Laboratory	6.25
*C. glabrata* Be10183	Clinical isolate	6.25
*C. glabrata* Be184	Clinical isolate	6.25

MIC: Minimal Inhibition Concentration; PLL-EO: *Pistacia lentiscus* L. leaves essential oil. Growth control with distilled water, always confirmed the growth of all the microbial strains in the absence of PLL-EO.

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
