# Peer review of "The Pharmaceutical Ability of *Pistacia lentiscus* L. Leaves Essential Oil Against Periodontal Bacteria and *Candida* sp. and Its Anti-Inflammatory Potential"

_antibiotics, 2020, doi:10.3390/antibiotics9060281_

Round 1

Reviewer 1 Report

I read your well-written manuscript with interest.

There are not many suggestions from my side to improve your manuscript.

However, consider inserting the reference "J N J Dent Assoc 2017 Mar;88(1):24-25" after the sentence "...other risk factors such as smoking, hyperglycaemia and, xerostomia."

Other than that, the manuscript is looking sharp.

Author Response

Dear Reviewer,

many thanks for appreciating our work and suggesting to support it with additional references. Regarding this point, however, we just have a long list. So, we would not like to follow the suggestion.

Best regards

Reviewer 2 Report

The manuscript "The pharmaceutical ability of Pistacia lentiscus L. leaves essential oil against periodontal bacteria and Candida sp. and its anti-inflammatory potential" is a very interesting and well prepared multicentre experimental research.

Given that herbal medicine and natural remedies are coming back now, as alternative antimicrobial and antifungal agents, the work has a large clinical aspect especially for dentists and ENT specialists. What is more, very widely used in clinical dentistry, including periodontology, preparations based on chlorhexidine, highly recommended by pharmaceutical companies, show cytotoxicity towards human fibroblasts, osteoblasts, which was particularly emphasized ina very good introduction.

Abstract: according to the guidelines of the Journal contains the key information.

Material and methods:  the research protocol is clearly and accurately presented, there is generally no comments except for the one question: why was October selected for leaf harvest? If that mattered, it would be worth explaining. The purpose of the work is clearly stated, however one could formulate a research hypothesis.

Results:  presented in 2 tables and 5 figures are well understood. An analysis of the chemical composition of the Pistacia lentiscus essential oil from leaves collected in the territory of Costa Paradiso-Sardinia was carried out. Have there been differences in the composition of this oil, based on the literature, for a plant harvested in another geographical region? If so, could these differences affect for example the genetic characteristics of the plant, its antioxidant capacity, and others, and thus its therapeutic properties? 

Discussion - very well-read discussion, however due to the clinically extremely important test results:  confirmation of the lack of PLL-EO cytotoxicity, in the discussion chapter, in my opinion, despite the fact that it is in vitro research, several sentences related to the use of the oil in clinical procedures were missing; despite the fact that the conclusions included the following sentence: p.145 ........"should indicate this oil as a natural component of safe therapeutic compounds as an alternative to chemicals.

Author Response

Dear Reviewer,

First of all, many thanks for appreciating our work and suggesting us how to increase it strength.

Following, our replies to your kind request of explanations and suggestions: 

Material and methods

1) the research protocol is clearly and accurately presented, there is generally no comments except for the one question: why was October selected for leaf harvest? If that mattered, it would be worth explaining. 

Thank you for this interesting question. 

The leaves of P. lentiscus were harvested in October according to the fact that it is an evergreen shrub that under Mediterranean climate conditions is subjected to a severe leaf senescence occurring during summer time with a peak of leaf abscission between July and August (Munné S. and Peñuelas J., 2003). While young leaves become predominant and mature starting from October. Harvesting at this time of the year allows attaining the highest amount of young-mature metabolic active leaves, before the winter senescence takes place. Concerning metabolic activity, the same occurs in the early spring, but leaves are elder but still not senescing.

Munné S. and Peñuelas J. 2003. Photo and antioxidative protection during summer leaf senescence in Pistacia lentiscus L. under Mediterranean field conditions. Annals of Botany. 92: 385-391.

We have explained this point in lines 308-310. 

2) The purpose of the work is clearly stated, however one could formulate a research hypothesis.

Applicable. We added the required hypothesis in line 100 after the “Aim” of the study as following:

The hypothesis of the present study was that PLL-EO has anti-microbial and anti-inflammatory activities without negatively interfering to human cells viability.

Results 

Have there been differences in the composition of this oil, based on the literature, for a plant harvested in another geographical region? If so, could these differences affect for example the genetic characteristics of the plant, its antioxidant capacity, and others, and thus its therapeutic properties? 

We thanks the reviewer for this interesting question. It is a fact that chemical composition of PLL-EO depends on the geographical region as it is strongly influenced by the environment of growing, which has an impact on the genetic character and then, on the type and concentrations of secondary metabolites in its fingerprint (Barra et al 2007). Thus the therapeutic properties between the oils may be a little bit different as well. Our PLL-EO, collected in the North of Sardinia, Costa Paradiso, was higher in α-pinene, and terpinen-4-ol content when comparing to that collected in Turkey [Duru et al 2003], and Iran (Allaith et al 2008). All these terpen have been indicated as bioactive terpene rich in medicinal properties (Rivas da Silva et al 2012). Thus, our data are different in comparison of those of other reports, which have investigated on oils of PLL leaves from plants growing in different Counties. 

Regarding the antioxidant capacity of different PLL essential oils, different results can be explained also by the period of harvesting and the method used to quantify them (for instance the spectrometric DPPH assay or the electron chemical assay).

Barra A., Coroneo V., Dessì S., Cabras P., Angioni A. Characterization of the volatile constituents in the essential oil of Pistacia lentiscus L. from different origins and its antifungal and antioxidant activity.  J Agric Food Chem. 2007, 22, 7093-7098.

Duru M.E, Cakir A., Kordali S., Zengin H., Harmandar M., Izumi S., Hirata T. Chemical composition and antifungal properties of essential oils of three Pistacia species. Fitoterapia. 2003, 74, 170-176.

Allaith S.A., Alfekaik D.F. Alssirag M.A. Identification of Pistacia vera and Prunus amygdalus Batsch seed oils using GC-MS as useful methodology for chemical classification. Food Chemistry 107 (2008) 1120–1130

Discussion - very well-read discussion, however due to the clinically extremely important test results:  confirmation of the lack of PLL-EO cytotoxicity, in the discussion chapter, in my opinion, despite the fact that it is in vitro research, several sentences related to the use of the oil in clinical procedures were missing; despite the fact that the conclusions included the following sentence: p.145 ........"should indicate this oil as a natural component of safe therapeutic compounds as an alternative to chemicals.

Thanks you for giving us the possibility to emphasize the biocompatibility of our PLL-EO and suggest a possible use as an antiseptic. We have added in the Discussion section the points you have kindly requested at the lines 208-212; and 223-227. Then, we have modified the Discussion section at the lines 243-256; and then the Conclusion section at the lines 438-442.

Many thanks again for your review.

Best regards